# Activities for Residents of Dutch Nursing Homes during the COVID-19 Pandemic: A Qualitative Study

**DOI:** 10.3390/ijerph19095465

**Published:** 2022-04-30

**Authors:** Marlon M. P. Smeitink, Hanneke J. A. Smaling, Lisa S. van Tol, Miriam L. Haaksma, Monique A. A. Caljouw, Wilco P. Achterberg

**Affiliations:** 1University Network for the Care Sector Zuid-Holland, Leiden University Medical Center, 2333 ZA Leiden, The Netherlands; h.j.a.smaling@lumc.nl (H.J.A.S.); l.s.van_tol@lumc.nl (L.S.v.T.); m.l.haaksma@lumc.nl (M.L.H.); m.a.a.caljouw@lumc.nl (M.A.A.C.); w.p.achterberg@lumc.nl (W.P.A.); 2Department of Public Health and Primary Care, Leiden University Medical Center, 2333 ZA Leiden, The Netherlands

**Keywords:** meaningful activities, geriatric care, dementia, nursing home, long-term care, SARS-Cov-2, COVID-19, pandemic

## Abstract

To protect nursing home residents from getting infected with COVID-19, several measures have been imposed. The aim of this study was to describe the impact of these measures on activities for Dutch nursing home residents, the conditions under which the activities could take place, and the considerations when making decisions about the (dis)continuation of activities. The study consisted of the data of the qualitative MINUTES-study. Textual units derived from documentation of an outbreak team (OT) meetings on activities, well-being, informal caregivers, and volunteers from 39 long-term care organizations were re-analyzed using a content analysis. The results shows that OTs more often discussed restarting and continuing activities than stopping activities during the COVID-19 pandemic. There were differences between time periods, but activities never completely stopped according to the minutes. Activities were offered in an adapted way, often under certain conditions, such as organizing activities at other locations (e.g., outside), with limited group size, and following specific guidelines. The main focus of the considerations made were the ability to adhere to the guidelines, the well-being of residents, ensuring safety, and balancing benefits versus risks given vaccination availability and coverage. Overall, the study showed that organizing activities for nursing home residents despite COVID-19 measures is possible.

## 1. Introduction

In December 2019, the first person was infected by SARS-Cov-2, which has spread over the world rapidly since [1]. The World Health Organization (WHO) officially announced the coronavirus pandemic in March 2020 [2]. The first symptomatic patient with a COVID-19 infection was reported in the Netherlands on 27 February 2020 [3]. Within four weeks, the Dutch government imposed a national lockdown. This first lockdown ended on 11 May 2020 and was later known as the first wave [4]. By the end of September 2020, the second COVID-19 wave started [5]. This wave led to a new lockdown from 14 October 2020 until 5 June 2021 [6,7].

Infections were reported all over the world [8], and Europe seemed to be the worst-infected continent according to available data [9]. By October 2021, in the Netherlands, a total of 48,570 people from long-term care (LTC) facilities were reported infected with the coronavirus. This is 38.9% of the approximal 125,000 residents of LTC facilities in the Netherlands and 2.1% (2,295,107 in total) of the reported infections [10,11,12]. To protect nursing home residents from getting infected, several regulations and restrictions have been imposed.

During the first lockdown, similar to other countries around the world [13,14], the 1.5 m distance was implemented, and a visitors ban was imposed in the Netherlands from 20 March 2020 until the end of May 2020 [15]. Residents were not allowed to leave the property or, when infected, their room [14,16,17,18]. Furthermore, worldwide but also in the Netherlands, all social activities and group activities were canceled or adjusted to the measures in force [13,17,19,20].

During the second lockdown, the visitor ban was no longer imposed although infected persons still had to isolate themselves, and the 1.5 m distance was maintained. The Netherlands was the only country that mandated that nursing homes reopen to visitors after research showed that this did not cause a rise in infections [15].

The visit ban and the change in organized activities had an impact on challenging behavior in nursing home residents [14,19,21]. Healthcare professionals were positive about the adjusted activities, such as small-scale activities and person-oriented activities, because they had a calming effect according to therapists [22]. Besides, a decrease in psychotic behavior and agitated behavior was noted, while depressive behavior and apathetic behavior increased [22]. One of the influencing factors was a diagnosis of dementia and the associated stage of the syndrome [19,21]. Specifically, residents with mild to moderate cognitive impairments seemed to be more severely impacted by the COVID-19 regulations [23].

There is ample literature indicating that a lack of (meaningful) activities during the COVID-19-pandemic has a negative impact on mental health [24,25,26,27,28,29]. However, most of those studies focused on community-dwelling older persons or those living within an assisted-living setting. Less is known about the impact of the COVID-19 pandemic on activities for nursing home residents and the challenges experienced by nursing homes regarding those activities given the government restrictions. One of the few studies about activities for nursing home residents during the COVID-19 pandemic showed that residents that participated in recreational activities had significantly better mental health [30].

The aim of this study was to describe the impact of the COVID-19 measures on the organization of activities for residents of Dutch nursing homes. First, an overview of the activities that were cancelled, started, and continued is given. Second, the differences in (dis)continuation of activities between the COVID-19 waves are described. Finally, we examined which factors were considered when making decisions on the (dis)continuation of activities for nursing home residents.

## 2. Materials and Methods

For the present qualitative study, data of the COVID-19 MINUTES study [31], a large Dutch multi-center study in which 41 LTC organizations shared the minutes of their outbreak team (OT) meetings on a weekly basis with the researchers, was used. The data were collected from week 10, 2020 until week 41, 2021. The recruitment and study procedures are described in more detail elsewhere [31].

For this study, the textual units coded as *activities*, *well-being, informal caregivers* and *volunteers* were selected from the MINUTES study dataset and recoded [31]. First, relevant textual units were selected. Textual units were included when about activities, conditions, or considerations within the LTC setting. This study defines *activities* as all the organized leisure activities without a therapeutic aim. Examples are making music together, exercising, and going outdoors. Drinking tea or eating dinner together is considered as a leisure activity because it is a social event. Personal body care (e.g., going to a hairdresser or pedicure) was not considered an activity in this study because they are part of daily care. Well-being is used to express all forms of well-being, such as physical, mental, and social well-being. The conditions contained requirements that needed to be met in order to organize a certain activity. Textual units were excluded when focused on activities with a therapeutic aim, on home care, on informal caregivers and volunteers without mentioning activities, or being unclear.

Next, textual units were divided in four periods: the first wave from week 10, 2020 until week 19, 2020 (T1); the intermediate period from week 20, 2020 until week 38, 2020 (T2); the second and third wave from week 39, 2020 until week 6, 2021 (T3); and vaccination period from week 7, 2021 until week 41, 2021 (T4). A content analysis [32] was used to re-analyze the selected textual units in this particular study because the existing theory and literature on this topic was limited.

Two researchers (M.S., H.S.) independently coded 75 randomly selected textual units per period and composed an initial list of codes. During a consensus meeting, the initial codebook was composed. Next, the 300 textual units were re-coded. During a second consensus meeting, discrepancies were discussed. Then, the remaining textual units were coded. During a third meeting, differences were discussed, and consensus was reached. A few codes were added to the codebook. The final codebook consists of seven categories with 48 unique topics in total.

## 3. Results

The complete dataset of the MINUTES study consisted of 11,711 textual units (Figure 1). Out of those units, 884 (7.5%) were coded as *Activities, Well-being, Informal Caregivers* or *Volunteers*. Another 431 units were excluded that did not meet the inclusions criteria, resulting in a total of 453 units that were analyzed. Two of the forty-one LTC organizations did not report on the study topic.

An overview of the frequency of the mentioned codes about activities per wave is given in Figure 2 and Appendix A. The most textual units were coded as restarted activities (*n* = 160), continued activities (*n* = 153), and stopped activities (*n* = 70). Some textual units included an activity whereof the start, stop, or continuation was unknown (*n* = 41). Moreover, 272 textual units were mentioned, and 85 contained considerations.

### 3.1. Stopped, Restarted, and Continued Activities over Time

During the first wave, the numbers of textual units coded as stopped, restarted, or continued activities were all between 12 and 23 textual units, as shown in Appendix A. During the intermediate period, most textual units were about restarting activities, and in the second and third wave, continuation of activities was mostly discussed. The vaccination period included most textual units about restarting and continuation of activities, mainly due to the reopening of restaurants: “*National policy is: terraces open as of April 28* […]. *The terraces of the nursing homes of* [the organization] *will be open as of April 28. The tables are placed in such a way that there is sufficient distance. Up to four persons can sit at each table at 1.5 m distance. Terraces are open for residents, visitors and staff*” (XB, week 17, 2021).

#### 3.1.1. Stopped Activities

Stopping activities were mostly mentioned by OTs during the intermediate period. Of all stopped activities that were mentioned, most were about restaurants, going on daytrips, and singing activities. Singing was cancelled during all waves: “*Singing is not allowed for the time being because the risk of contamination is higher with this activity*” (XZ, week 27, 2020). Exercising, going outside, religious activities, memorial services, and music activities were almost never mentioned as stopped.

During the first wave, activities that OTs discussed to stop were group activities, going outside, and visits by family. Furthermore, daycare centers were mentioned as closed, and family visits were frequently prohibited. During the intermediate period, OTs discussed to stop day trips, such as visiting museums and visiting family outside the nursing home. Moreover, pets were not allowed: “*Assistance dogs: (in general) no animals allowed at the locations for the time being*” (XF, week 27, 2020). During the second and third wave, the closing of restaurants was mostly mentioned. The vaccination period and first wave had the least stopped activities mentioned.

#### 3.1.2. Restarted Activities

Most activities were mentioned as restarted in the intermediate period. In the intermediate period, restaurants re-opened only for staff and residents, with a maximum group size of 30 persons, while during the vaccination period, restaurants also opened for external guests following the national guidelines. Appendix A shows that most textual units about restarted activities included daycare centers, restaurants, exercising, and going outside.

During the last three weeks (week 17 until week 19, 2020) of the first wave, most daycare centers restarted although with several adjustments: “*Search for locations for starting up daycare centers. Not in our own nursing homes, check whether municipalities have locations available for daycare centers*” (XM, week 19, 2020) “*Daycare centers are slowly being restarted in small groups at all nursing home locations* […]” (YG, week 19, 2020). This restarting of activities in daycare centers continued in the first weeks (until week 25) of the intermediate period. However, some organizations only restarted daycare centers from week 27, 2020 (intermediate period).

#### 3.1.3. Continued Activities

During the first wave, daycare centers remained open in some organizations, while they closed in others. The intermediate period was characterized by scaling-up activities. Residents went outside, and exercise and music activities were still offered. Organizations tried to continue the holiday festivities and keep the restaurants open for residents. These were the most often discussed activities during the second and third wave: “*Within residential care there is an explicit wish to allow residents to eat meals together. Residents need this and lose weight, because they are currently eating less*” (XQ, week 3, 2021). The continuation of restaurants was also an often-discussed topic in the vaccination period.

Organizations tried to keep their activities going as long as possible: “*The joint activities in the living rooms will be maintained for as long as possible- outside the living rooms all joint activities will be stopped*” (XR, week 12, 2020). The daycare center and the facilities’ shops were also kept open as long as feasible. The parameters of activities (e.g., location or group size) were changed by organizations to facilitate the continuation of the activities within the restrictions and to follow government guidelines. Of the activities that continued, most textual units were about the upscaling of activities and downscaling of COVID-19 measures, holidays, daycare centers, and opening restaurants.

### 3.2. Conditions and Considerations and the Difference per Time Period

Conditions and considerations for decisions about continuing, stopping, and (re)starting activities were also discussed by the OTs. Table 1 presents an overview of the mentioned conditions and considerations per time period.

#### 3.2.1. Conditions and the Difference per Time Period

Out of all the conditions, following the guidelines of the government was mentioned the most. Group size, location, without family/external visitors, and not having any health-related problems were also mentioned often.

During the first wave, organizing activities at other locations (e.g., outside or other external locations) and having no health-related COVID-19 symptoms were the most frequently mentioned conditions. Going outside and going to the daycare centers under the condition that someone had no health-related COVID-19 symptoms were mostly mentioned: “*Walking outside with family, provided that family has no COVID-19-related symptoms, is allowed*” (YA, week 12, 2020).

The intermediate period contained the most conditions about following the guidelines of the government. Guidelines were mentioned in general or in more specific detail: “*The recreational bus may be used again provided that the guidelines are followed. These include performing a health check and wearing a face mask*” (XX, week 30, 2020); “*Residents with decision-making capacity who can adhere to the 1.5 m guideline are allowed to go outside independently.” (XO, week 21, 2020);* “*Attention must be paid to ventilation and the maximum number of people in a room during activities. Organize more activities in the restaurant if necessary. The ventilation plan must be linked to activities at a location*” (YF, week 40, 2020).

The vaccination rate within the organization was mentioned as a requirement for organizing and participating in activities: “*Only location X has to wait a little longer* [until activities will be organized again] *until the residents there have been vaccinated*” (XZ, week 13, 2021).

#### 3.2.2. Considerations and the Difference in Time

The considerations regarding the (dis)continuation of activities made by the OTs during the different time periods are shown in Table 1. The main focus of those considerations were the ability to adhere to the guidelines of the government, the well-being of residents, ensuring safety, and balancing benefits versus risks of COVID-19 given vaccination availability and coverage. Most of those considerations were mentioned in the second and third wave and in the vaccination period. The vaccination period included all considerations that centered around vaccinations.

Considerations about the ability to adhere to the guidelines were mostly made during the second and third wave and the vaccination period. Residents’ well-being was also most often discussed during the second and third wave. Organizations looked for ways to work with the guidelines while looking out for the well-being of residents: “*Looking for loopholes to not having to close the restaurants for the social well-being of our residents*” (XN, week 42, 2020); “*Propose alternatives for the week-start and church service, which are in line with national guidelines and do justice to the needs of residents as much as possible*” (XE, week 5, 2021).

The safety of residents and balancing the benefits of a choice for a certain measure versus the danger of not taking this measure was also most often discussed during the fourth period (T4): “*The question has raised whether decorative chicks can be placed in the hallway around Easter. Due to the risk of groups forming around the chicks, it is decided not to do that this year*” (YI, week 9, 2021).

## 4. Discussion

The aim of this study was to describe the impact of the COVID-19 measures on the organization of activities for residents of Dutch nursing homes. Our overview of the activities for nursing home residents indicated that, in general, OTs more often discussed continuing than stopping activities during the COVID-19 pandemic. When activities restarted and continued, this was often under conditions such as using external locations, limited group sizes, and applying specific guidelines. The reduction of measures, guidelines, vaccinations, and safety were most discussed while considering the restart, continuation, or stop of activities.

The most textual units about continued activities were mentioned during the second and third wave, when almost none were mentioned as restarted. This corresponds to earlier research findings on data of September and October 2020 that reported that residents could attend activities again in 59 Dutch nursing homes in October 2020 [33]. Although previous findings of studies that collected data between April and June 2020 [19,20] showed that most activities for nursing home residents were mentioned as stopped, our findings suggest that more activities restarted or continued in every time period. This shows that our large nursing home sample was able to retain activities despite the difficult situation during the pandemic.

In addition to differences between time periods, there were also differences within time periods. A proper example is daycare centers. OTs mentioned both the closing and continuation of daycare centers the most during the first wave. This can be explained by the different phases within one time period. During the first period of the first wave, daycare centers remained open as long as possible, but when infection rates skyrocketed, and mortality rates increased during the second part of wave one, daycare centers had to close to prevent further spread of the virus.

This study has several strengths. One of them is the broad insight it provides into the impact of the measures on activities in nursing homes nationwide. The study gives insight into the considerations that were made regarding the (dis)continuation of the activities for the nursing home residents as well. These rich data were collected without placing an extra burden on nursing homes for an extended period of time, enabling us to make comparisons between time periods during the COVID pandemic [31]. Besides, the data were collected during the pandemic instead of afterwards, which gives a direct insight in the decision-making process instead of a retrospective view of participants on the topic.

Limitations of the study are the lack of context in which decisions and considerations were made. The textual units were often concrete, short, and without many details. Moreover, the stop, start or continuation of activities could have been discussed in other meetings instead of the OT meeting. Therefore, it is only possible to make conclusions about the (dis)continuation of the specific activities that were mentioned in the minutes. This limits the generalizability of our data.

The relatively small number of textual units in the MINUTES study dedicated by the OTs to *activities*, *volunteers, informal caregivers,* and *well-being,* is striking considering the principles of person-centered care of the National Institute for Health and Care Excellence UK (NICE) and the Dutch quality framework for nursing homes [34,35]. Both highlight that attention should be paid to the well-being of nursing home residents with dementia and the organization of (individualized) activities. The relatively low amount of textual units suggests that the focus of the OTs was less on the well-being of residents and organized activities but rather on how to stop infections while still being able to provide the daily care. Although this study showed that activities continued, more research on the considerations of long-term care organizations on decisions about the (dis)continuation of activities outside the OTs is necessary, as is research focused on disentangling the relationship between residents’ well-being, (meaningful) activities, and the restrictive measures.

## 5. Conclusions

This study showed that it is possible to organize activities for nursing home residents despite COVID-19-related measures although some creativity is needed and important to meet the restrictions and safety regulations. This is an important finding considering the significance of activities for the well-being and quality of life of nursing home residents. If there is one thing that the pandemic has taught long-term care, it is that we should always strive for a good balance between well-being and safety, and in the beginning of the pandemic, the scale was tilted too much towards safety. Overall, the study showed that nursing homes have put more energy into keeping activities accessible than earlier described.

## Figures and Tables

**Figure 1 ijerph-19-05465-f001:**
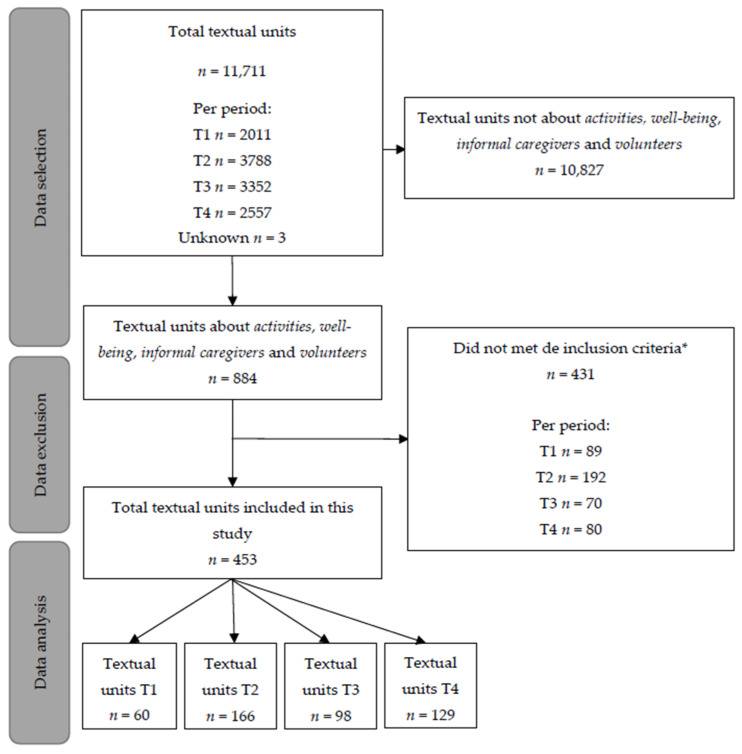
Flowchart of the included textual units per time period. Note: T1 is the first wave, T2 is the intermediate period, T3 is the second and third wave, and T4 is the vaccination period. * Exclusion criteria were: activities with a therapeutic aim, not in long-term care, concerning family caregivers and volunteers without mentioning activities, or were unclear.

**Figure 2 ijerph-19-05465-f002:**
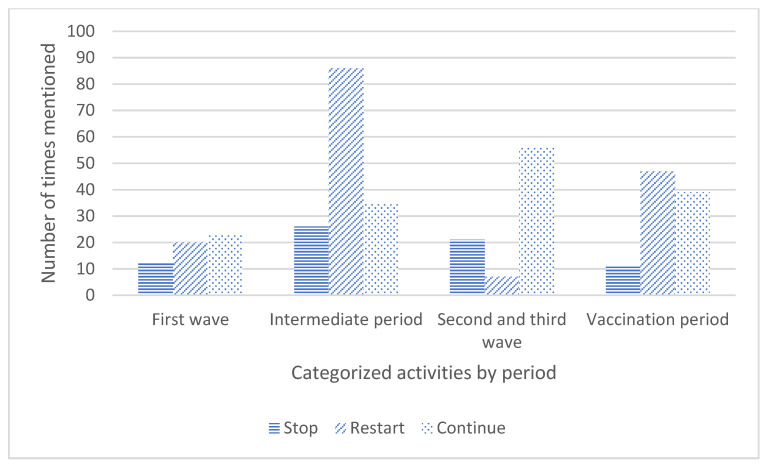
Overview of stopped, restarted, and continued activities for nursing home residents per time period.

**Table 1 ijerph-19-05465-t001:** Overview of the mentioned conditions and considerations per time period.

		First Wave	Intermediate Period	Second and Third Wave	Vaccination Period
Conditions					
	Condition not specified	-	1	1	-
	Access via non-infected wards ^1^	1	3	-	1
	Limited contact with the outside world exception for close family	2	3	1	1
	National guidelines must be followed	3	39	17	38
	Only allowed when negative for COVID-19 (family, resident, staff) or no COVID-19-related symptoms	6	4	4	4
	Only allowed when residents suffer from loneliness or being sad	-	-	-	1
	Only allowed with a negative COVID-19 (self-)test	-	-	-	1
	Organized at a specific location (e.g., outside), online	9	16	13	2
	Resident needs to be independent or have decision-making capacity	-	6	-	-
	Residents are in need of activities	1	1	-	2
	Restriction in group size	3	17	11	13
	Scaling down guidelines for activities	-	-	-	5
	Transport to and from activities (no group transport)	1	1	-	-
	When (fully) vaccinated	-	-	-	7
	Without pets	1	-	-	1
	Without singing, at a safe distance, no choir	-	7	3	3
	Without visitors, family or volunteers	-	-	11	7
Considerations					
	Consideration not specified	1	1	-	1
	Active dying phase	1	-	-	-
	Attune to safety	1	2	4	10
	Enforcement options	-	-	-	1
	Guidelines	2	-	-	1
	Health risks and health benefits	1	3	2	1
	Holidays	-	-	2	-
	Importance of well-being of residents	-	5	9	4
	Limit contact with the outside world	-	-	2	1
	Scaling up activities and group size	-	1	-	-
	Unable to adhere to the guidelines and imposed restrictions	-	2	8	8
	Vaccination rate	-	-	-	11

Note: ^1^ Prescribed route to activity location; otherwise, the activities are not allowed to continue. -, there were no textual units coded with this code in this category in the time selected period

## Data Availability

The data presented in this study are available on request from the corresponding author. The data are not publicly available due to the agreement with participating organizations. During the consent process, participating organizations were explicitly guaranteed that the data would be pseudonymized by the study’s research center and that pseudonymized data would only be seen by members of the study team. For any discussions about the dataset, please contact UNC-ZH@lumc.nl.

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
