# Peer review of "Activities for Residents of Dutch Nursing Homes during the COVID-19 Pandemic: A Qualitative Study"

_ijerph, 2022, doi:10.3390/ijerph19095465_

Round 1
Reviewer 1 Report
Thank you for the opportunity to review this paper. It is an interesting study of available data that gives insight into the decisions made about activities in nursing homes during the pandemic. Most of the comments I have to make are small adjustments to English grammar, and clarification in wording to be sure you are communicating what you intend.
Line 37 – Is this to say that between the beginning of the pandemic and October 2021, 48,570 LTC residents had been infected? If so “By October 2021” would communicate that better.
Lind 49 – “Rise” rather than “raise”
Line 57 – remove “a”
Line 136 – I’m curious about this, since religious activities, memorial services and music activities were stopped in many Canadian nursing homes (or moved to virtual platforms, or very small groups). In line 138 “group activities” were stopped – Does this not include gathering for religious activities, memorial services and music activities? I wonder if the fact that they were not mentioned as being stopped means that they continued, or not. I would be tempted as a researcher of spiritual care, to cite this paper as evidence that, in Dutch nursing homes, religious activities continued. But do you think that is actually true?? If not, I would suggest adding a line to qualify this, to prevent researchers like me from misinterpreting your data.
Line 65/67 – “was used” is repeated – remove one.
Line 141 – visitING family
Line 143/144 – repeat of “during the second and third wave” – remove one
Line 242 – Do you mean “retain” rather than “pertain”?
Line 253 – I don’t understand the word “stadia”
Line 257 and 259 – insight “into” rather than “in”
Line 286 - “taught” rather than “learned”
Line 288 – the scale was TILTED too much…
Line 289 – “into” rather than “in”, and “than” rather than “then”
Thanks again for this paper. I believe it is a meaningful contribution to the literature on experience in nursing homes during the pandemic.
Author Response
Reviewer report 1
- Most of the comments I have to make are small adjustments to English grammar, and clarification in wording to be sure you are communicating what you intend.
We would like to thank the reviewer for their grammar suggestions. We have adopted all suggested grammar adjustments.
- I’m curious about this, since religious activities, memorial services and music activities were stopped in many Canadian nursing homes (or moved to virtual platforms, or very small groups). In line 138 “group activities” were stopped – Does this not include gathering for religious activities, memorial services and music activities? I wonder if the fact that they were not mentioned as being stopped means that they continued, or not. I would be tempted as a researcher of spiritual care, to cite this paper as evidence that, in Dutch nursing homes, religious activities continued. But do you think that is actually true?? If not, I would suggest adding a line to qualify this, to prevent researchers like me from misinterpreting your data.
It is important to note that we can only draw conclusions about the activities which were mentioned as stopped, continued or restarted in the outbreak team minutes. As we note in the discussion (line 280-282: Besides, the stop, start or continuation of activities could have been discussed in other meetings instead of the OT meeting), the (dis)continuation of activities could also have been discussed outside the outbreak team meetings. Therefore, stating that religious activities continued is probably too bold. We added line 283-284 to emphasize this (Line 283-284: Therefore, it is only possible to conclude something about the (dis)continuation of the specific activities which were mentioned in the minutes).
Besides, the reviewer mentions the code ‘’group activities’’. This code refers to textual units that mention group activities in general instead of specific activities. It is indeed possible that those group activities included religious activities, memorial services and music activities, but we cannot be sure. Appendix 1 shows more specific information about the time period in which music activities, religious activities and memorials services (dis)continued.

Reviewer 2 Report
The manuscript consists of total 11 pages, including 2 figures and 2 tables, and the list of total 26 literature references. The article presents the results of the original study concerning the spectrum of activities that were stopped, continued or started by the residents of Dutch nursing homes in different phases of the COVID-19 pandemic. As such it fits into the topics spectrum of the Journal. The manuscript has logical structure and is written in good quality English.
The title of the article adequately represents its contents.
The Abstract section properly mirrors both the structure and the contents of the main text.
The Introduction provides enough information on the background of the researched problem.
The Material and Methods section presents the chosen methodology in enough detail.
The Results section is presenting the information derived from the study that is consistent with the chosen methodology in enough details, the table in the appendix is very useful for understanding the results so I would suggest that the Authors consider making it element of the main text instead of putting it into appendix.
The Discussion section places the study results into the broader context as well as provides the information on the perceived strengths and weaknesses of the study.
The Conclusions are based on presented and discussed results.
The Authors may consider adding mentions of the following aspects to broaden the background of the presented problem:
- the experiences and limitations of LTC during COVID-19 in different countries and its perceived results, e.c. as in:
https://doi.org/10.3390/ijerph19074275
https://doi.org/10.3390/epidemiologia3020014
https://doi.org/10.3390/ijerph19010519
https://doi.org/10.3390/brainsci12010062
https://doi.org/10.3390/ijerph19010075
https://doi.org/10.3390/ijerph19073836
https://doi.org/10.3390/brainsci11080986
https://doi.org/10.3390/ijerph181910099
https://doi.org/10.3390/ijerph19042252
Author Response
Reviewer report 2
The manuscript consists of total 11 pages, including 2 figures and 2 tables, and the list of total 26 literature references. The article presents the results of the original study concerning the spectrum of activities that were stopped, continued or started by the residents of Dutch nursing homes in different phases of the COVID-19 pandemic. As such it fits into the topics spectrum of the Journal. The manuscript has logical structure and is written in good quality English.
- The table in the appendix is very useful for understanding the results so I would suggest that the Authors consider making it element of the main text instead of putting it into appendix.
To emphasize the qualitative aim of this study, we decided to keep this Table as Appendix A instead of in the main text.
- The Authors may consider adding mentions of the following aspects to broaden the background of the presented problem:
- the experiences and limitations of LTC during COVID-19 in different countries and its perceived results, e.c. as in [9 literature suggestions].
We would like to thank the reviewer for the suggestion to add the experiences and limitations of different countries and emphasizes this with literature suggestions. We included several of the suggested references to point out the COVID-19 experiences and limitations worldwide. For example:
- Line 37-38: Infections were reported all over the world and Europe seemed to be the worst infected continent according to available data.
- Line 44-46: During the first lockdown, similar to other countries around the world, the 1.5 meter distance was implemented and a visitors ban was imposed in the Netherlands from March 20, 2020 until the end of May 2020.
- Line 47-49: Furthermore, worldwide, but also in the Netherlands, all social activities and group activities were canceled or adjusted to the measures in force.

Reviewer 3 Report
Given my passion for qualitative research and qualitative analysis, I read the article with great interest. And I must admit that the initial interest was confirmed by a clean and essential style of writing.
The simple, essential and clearly definitive objectives.
However, I would like to ask the authors for a small effort to clarify some points.
First of all, the methodological approach. The reference to the article by Hsieh and Shannon (2005) gives a general explanation. However, this is an article published 17 years ago and in these 17 years the world of qualitative analysis has changed, it has also evolved. For example, I would see well in the quantitative tables I would see a more "narrative" description of the data, a sort of "storyline" describing the different pandemic phases or a storyline for each period.
Similarly, to give a greater sense to the "qualitative" data, I ask the authors if they have tried to make a Wordcloud on the raw data to verify if the key words of the project (activities, well-being and volunteers) are equally represented.
Without arriving at sophisticated computer-assisted analyzes (for example with MAXQDA or Nvivo), I think that the authors' approach can be made in a more current and modern key.
In general (and it is certainly not the Authors' fault) I am always perplexed when a qualitative research shows tables full of numbers, even if they are simple frequencies. I believe that those who do qualitative research can render a better service to their data.
for the same reason, I would avoid direct references to terms such as "impact" (line 233 and abstract) which follow a deterministic approach that is not typical of qualitative research.
Finally, pay close attention to the MDPI editorial rules for the bibliography because there is a lot of discrepancy. For example, the year of publication is bold in some cases (lines 366 and 363) and not in others; Doi is underlined in some cases (line 396) and not in others, magazine titles are shortened in some cases (375) and not in others; titles are capitalized on line 359-360 and not in other cases. In line 331, the month (februari) is indicated which perhaps is not required and in any case is not in English.
Details, but denote careful writing.
Author Response
Reviewer report 3
Given my passion for qualitative research and qualitative analysis, I read the article with great interest. And I must admit that the initial interest was confirmed by a clean and essential style of writing.
The simple, essential and clearly definitive objectives. However, I would like to ask the authors for a small effort to clarify some points.
- First of all, the methodological approach. The reference to the article by Hsieh and Shannon (2005) gives a general explanation. However, this is an article published 17 years ago and in these 17 years the world of qualitative analysis has changed, it has also evolved. For example, I would see well in the quantitative tables I would see a more "narrative" description of the data, a sort of "storyline" describing the different pandemic phases or a storyline for each period.
We replaced the reference of Hsieh and Shannon (2005) by a more recent reference: https://doi.org/10.1177/1049732317753586 in line 99.
Regarding the description of our data, it should be noted that the study contains a lot of textual units whereof we wanted to give a broad overview. The data were weekly collected across a time span of 20 months. This produced a lot of data (textual units). Therefore, we decided to stick to describing the results per category while using the specific activities. While section 3.2 has a more chronological narrative, we did not decide to write 3.1 in this chronological order, because not every wave included data about the same activities. As a result it is difficult to compare activities between the different waves.
- Similarly, to give a greater sense to the "qualitative" data, I ask the authors if they have tried to make a Wordcloud on the raw data to verify if the key words of the project (activities, well-being and volunteers) are equally represented
We would like to thank the reviewer for their suggestion to make a wordcloud from the raw data to verify if the key words of the project are equally represented. However, we decided not to do this, as the main focus of the study is to examine the impact of COVID-19 measures on the organization of activities in Dutch nursing homes. The categories well-being, informal caregivers and volunteers were merely used to capture some additional textual units related to activities. We did not expect or aim to analyze an equal representation of these categories, as most textual units about activities were expected to be labeled as such. We expect that a wordcloud would have shown this.
For your information, we did look at the number of textual units per time period per category. The number outside the brackets is the number of textual units per code per time period in the raw data and the numbers between the brackets is the number of included textual units in this study:
|
|
Activities |
well-being |
volunteers |
Informal care |
|
T1 |
58 (41) |
31 (11) |
29 (3) |
31 (5) |
|
T2 |
178 (130) |
84 (31) |
74 (3) |
22 (2) |
|
T3 |
90 (73) |
58 (23) |
11 (1) |
9 (1) |
|
T4 |
103 (138) |
48 (22) |
22 (3) |
1 (1) |
- Without arriving at sophisticated computer-assisted analyzes (for example with MAXQDA or Nvivo), I think that the authors' approach can be made in a more current and modern key.
In general (and it is certainly not the Authors' fault) I am always perplexed when a qualitative research shows tables full of numbers, even if they are simple frequencies. I believe that those who do qualitative research can render a better service to their data.
We agree with the reviewer that Tables 1 and 2 are rather quantitative. However, for the present study we believe these tables are important because the frequencies indicate when and how often the topics were discussed.
- For the same reason, I would avoid direct references to terms such as "impact" (line 233 and abstract) which follow a deterministic approach that is not typical of qualitative research.
We agree with the reviewer that the word ‘impact’ is often associated with quantitative research and therefore we changed two more quantitatively formulated sentences to be more qualitatively formulated:
- Line 145: Stopping activities were mostly mentioned by OTs during the intermediate period.
- Line 161: Most activities were mentioned as restarted in the intermediate period.
Moreover, we specifically refrained from making any quantitative statements regarding impact in our manuscript.
- Finally, pay close attention to the MDPI editorial rules for the bibliography because there is a lot of discrepancy. For example, the year of publication is bold in some cases (lines 366 and 363) and not in others; Doi is underlined in some cases (line 396) and not in others, magazine titles are shortened in some cases (375) and not in others; titles are capitalized on line 359-360 and not in other cases. In line 331, the month (februari) is indicated which perhaps is not required and in any case is not in English.
Details, but denote careful writing.
We would like to thank the reviewer for their attention to the details. We have made several adjustments to the bibliography to make sure everything is according to the MDPI reference style.

Reviewer 4 Report
I admire the authors for taking this important issue in hand to study and explaining nicely in a smart scientific manner. Here are my comments which should be addressed and added before considering for acceptance.
Introduction: The introduction should be broaden to show the overall impact. Also, explaining the situation in during the first wave in total European region. This article should be cited for adding few lines on the fact.
https://doi.org/10.1016/j.arcmed.2020.05.015
The authors have cited an interesting article authored by Leontjevas, R; Knippenberg, I.A.H and colleagues (Reference number 13). However, another interesting article from the same group Knippenberg et al. wasn't cited in the literature review. I would suggest the authors to follow this article for explaining more on the challenging behavior in nursing homes during the COVID-19 pandemic in Netherlands from the healthcare professionals' aspect also to give an overall picture. Link of the article is:
https://doi.org/10.1186/s12877-022-02824-y
Other important references are missing which will enlarge and widen up the introduction for global readership.
https://doi.org/10.1007/s41999-021-00531-2
https://doi.org/10.1155/2020/8870249
Figure 2: I would suggest to use patterned bar graph. The colors would be difficult to distinguish in BW printout.
Author Response
Reviewer report 4
I admire the authors for taking this important issue in hand to study and explaining nicely in a smart scientific manner. Here are my comments which should be addressed and added before considering for acceptance.
- Introduction: The introduction should be broaden to show the overall impact. Also, explaining the situation in during the first wave in total European region. This article should be cited for adding few lines on the fact. https://doi.org/10.1016/j.arcmed.2020.05.015
We would like to thank the reviewer for the suggestion to broaden the scope of this article. We added the suggested article line 37-38 (Infections were reported all over the world and Europe seemed to be the worst infected continent according to available data). Besides, we broadened the scope of the article by mentioning the restrictions worldwide:
- Line 44-46: During the first lockdown, similar to other countries around the world, the 1.5 meter distance was implemented and a visitors ban was imposed in the Netherlands from March 20, 2020 until the end of May 2020.
- Line 47-48: Furthermore, worldwide all social activities and group activities were canceled or adjusted to the measures.
- The authors have cited an interesting article authored by Leontjevas, R; Knippenberg, I.A.H and colleagues (Reference number 13). However, another interesting article from the same group Knippenberg et al. wasn't cited in the literature review. I would suggest the authors to follow this article for explaining more on the challenging behavior in nursing homes during the COVID-19 pandemic in Netherlands from the healthcare professionals' aspect also to give an overall picture. Link of the article is: https://doi.org/10.1186/s12877-022-02824-y
This suggestion was also added to our manuscript. We added line 53-60 to address the challenging behavior in nursing homes and the perspective of the healthcare professionals:
The visit bans and the change in organized activities had an impact on challenging behavior in nursing home residents. Healthcare professionals were positive about the adjusted activities such as small-scale activities and person-oriented activities, because they had a calming affect according to therapists. Besides, a decrease in psychotic behavior and agitated behavior was noted, while depressive behavior and apathetic behavior increased. One of the influencing factors was a diagnosis of dementia and the associated stage of the syndrome. Especially residents with mild to moderate cognitive impairments seemed to be more severely impacted by the COVID-19 regulations.
- Other important references are missing which will enlarge and widen up the introduction for global readership. https://doi.org/10.1007/s41999-021-00531-2 and https://doi.org/10.1155/2020/8870249.
To widen up the introduction for global readership, we added several references and made sure that we also stated that restrictions and effects of those restrictions were not only seen in the Netherlands but worldwide. We therefore also added both references mentioned by the reviewer.
- Figure 2:I would suggest to use patterned bar graph. The colors would be difficult to distinguish in BW printout.
We changed the bar graph to a patterned bar graph.
